# Comparing Heterologous and Homologous COVID-19 Vaccination: A Longitudinal Study of Antibody Decay

**DOI:** 10.3390/v15051162

**Published:** 2023-05-13

**Authors:** Chiara Orlandi, Giuseppe Stefanetti, Simone Barocci, Gloria Buffi, Aurora Diotallevi, Ettore Rocchi, Marcello Ceccarelli, Sara Peluso, Daniela Vandini, Eugenio Carlotti, Mauro Magnani, Luca Galluzzi, Anna Casabianca

**Affiliations:** 1Department of Biomolecular Sciences, Section of Biochemistry and Biotechnology, University of Urbino Carlo Bo, Via Arco d’Augusto 2, 61032 Fano, PU, Italy; chiara.orlandi@uniurb.it (C.O.); giuseppe.stefanetti@uniurb.it (G.S.); g.buffi@campus.uniurb.it (G.B.); aurora.diotallevi@uniurb.it (A.D.); m.ceccarelli3@campus.uniurb.it (M.C.); mauro.magnani@uniurb.it (M.M.); luca.galluzzi@uniurb.it (L.G.); 2Laboratorio Covid, University of Urbino Carlo Bo, Via Arco d’Augusto 2, 61032 Fano, PU, Italy; 3Department of Clinical Pathology, Azienda Sanitaria Unica Regionale Marche Area Vasta 1 (ASUR Marche AV1), Viale Comandino 70, 61029 Urbino, PU, Italy; simone.barocci@sanita.marche.it (S.B.); daniela.vandini@sanita.marche.it (D.V.); 4Department of Medical and Surgical Sciences, University of Bologna, Via Massarenti 9, 40138 Bologna, BO, Italy; ettore.rocchi2@studio.unibo.it; 5Department of Physics and Astronomy, University of Bologna, Viale Berti Pichat 6/2, 40127 Bologna, BO, Italy; sara.peluso2@studio.unibo.it; 6Department of Prevention, ASUR Marche AV1, Viale Comandino 21, 61029 Urbino, PU, Italy; eugenio.carlotti@sanita.marche.it

**Keywords:** SARS-CoV-2, COVID-19, heterologous vaccination, anti-Spike IgG response, antibody decay

## Abstract

The humoral response after vaccination was evaluated in 1248 individuals who received different COVID-19 vaccine schedules. The study compared subjects primed with adenoviral ChAdOx1-S (ChAd) and boosted with BNT162b2 (BNT) mRNA vaccines (ChAd/BNT) to homologous dosing with BNT/BNT or ChAd/ChAd vaccines. Serum samples were collected at two, four and six months after vaccination, and anti-Spike IgG responses were determined. The heterologous vaccination induced a more robust immune response than the two homologous vaccinations. ChAd/BNT induced a stronger immune response than ChAd/ChAd at all time points, whereas the differences between ChAd/BNT and BNT/BNT decreased over time and were not significant at six months. Furthermore, the kinetic parameters associated with IgG decay were estimated by applying a first-order kinetics equation. ChAd/BNT vaccination was associated with the longest time of anti-S IgG negativization and with a slow decay of the titer over time. Finally, analyzing factors influencing the immune response by ANCOVA analysis, it was found that the vaccine schedule had a significant impact on both the IgG titer and kinetic parameters, and having a Body Mass Index (BMI) above the overweight threshold was associated with an impaired immune response. Overall, the heterologous ChAd/BNT vaccination may offer longer-lasting protection against SARS-CoV-2 than homologous vaccination strategies.

## 1. Introduction

COVID-19 vaccination has played a crucial role in controlling the pandemic by preventing severe illness caused by SARS-CoV-2 and improving clinical outcomes. Although homologous vaccination, which involves administering the same vaccine product for all doses in a vaccination series, has been the traditional approach, there is growing interest in heterologous schedules, which involve using different vaccine products in a series. Heterologous schedules may offer immunologic advantages to extend the breadth and longevity of protection provided by the currently available vaccines [1] while simplifying mass vaccination programs and supply management. Therefore, evaluating evidence of specific heterologous regimens can guide future COVID-19 boosting approaches.

We have recently reported the analysis of the humoral response of subjects from a population-based serological survey in the northern area of the Marche region (Italy) vaccinated with mRNA and adenoviral-vector SARS-CoV-2 vaccines [2]. Homologous vaccination with two doses of Pfizer-BioNTech BNT162b2 (BNT) or Oxford-AstraZeneca ChAdOx1-S nCoV-19 (ChAd) was compared to heterologous vaccination with ChAdOx1-S followed by BNT162b2. Our findings showed that, two months after the second dose, the heterologous vaccine schedule resulted in a significantly higher anti-Spike IgG response compared to the homologous vaccine schedules. We also investigated some factors (i.e., vaccine schedules, sex, age, smoking status, BMI) that could potentially influence the humoral response, showing that only the type of vaccine used significantly affected the antibody titer. Based on our results, we reasoned that using a heterologous vaccination schedule is a safe and effective way to enhance immunity against COVID-19.

In this study, we followed up by analyzing the humoral responses induced by the three groups of vaccination (BNT/BNT, ChAd/ChAd and ChAd/BNT) four and six months after the second dose. Furthermore, we analyzed the antibody decay associated with each vaccination strategy by applying a first-order kinetics equation [3]. The goal was to obtain insights into the kinetics of IgG persistence associated with the different immunization strategies. Additionally, we used ANCOVA analysis to examine how the three vaccine schedules were associated with dependent variables such as the observed IgG titers or the predicted kinetic variables. We adjusted for various clinical variables, including sex, age, smoking status and BMI, to minimize their potential confounding effects on the observed relationships.

## 2. Materials and Methods

### 2.1. Recruitment and Study Cohort Characteristics

The study participants (*n* = 1248) were recruited from professionally active healthcare workers (Azienda Sanitaria Unica Regionale—Area Vasta 1 (ASUR Marche AV1); *n* = 952) and university personnel (University of Urbino Carlo Bo (UNIURB); *n* = 296) vaccinated against COVID-19 between December 2020 and June 2021 in Urbino (PU), Italy. The subjects were followed up at two, four and six months after immunization with BNT/BNT, ChAd/ChAd or ChAd/BNT. Only two individuals, belonging to the BNT/BNT group (0.2%), provided information about a positive COVID-19 diagnosis before the follow-up (July 2020). The serum of the vaccinated subjects was analyzed two months (mean ± SD, 60 ± 3 days), four months (mean ± SD, 120 ± 3 days) and six months (mean ± SD, 180 ± 4 days) after the second dose. The characteristics of the three groups are reported in Table 1A. In order to perform a kinetic analysis, we also analyzed a subset of patients who completed a follow-up consisting of antibody determination at the three time points (two, four and six months) and whose serum samples showed a descending phase of the anti-S IgG level. Of the 1248 vaccinated subjects, 779 completed the post-vaccination follow-up (ASUR Marche AV1 (*n* = 585) and UNIURB (*n* = 184)). In addition, serum samples that did not show a descending phase of anti-S IgG concentration were excluded (ASUR Marche AV1 (*n* = 185) and UNIURB (*n* = 9)). Overall, a total number of 585 patients were analyzed for kinetic studies (ASUR Marche AV1 (*n* = 410) and UNIURB (*n* = 175)). This sub-cohort included 381 females (65%) and 204 males (35%). The median (range) age of the study group was 52 (25–72). The characteristics of this sub-cohort are reported in Table 1B.

### 2.2. Determination of Antibody Levels

Serum samples were tested for SARS-CoV-2 IgG antibodies with a limited storage time (less than 4 days) using the “LIAISON^®^ SARS-CoV-2 TrimericS IgG” Chemiluminescent Immunoassay (CLIA) kit, as previously described [2], at the Clinical Pathology Laboratory of the Urbino Hospital (ASUR Marche AV1). The assay shows a high sensitivity (98.7%) and specificity (99.5%) for the detection of anti-trimeric SARS-CoV-2 Spike protein IgG. This method has a high positive percent agreement (95% CI: 97.8–100.0%) and a negative rate of 96.9% (95%CI: 92.9–98.7%) with neutralizing IgG antibodies. The quantification range is between 4.81 and 2080 BAU/mL, and the cut-off for positivity is 33.8 BAU/mL (conversion factor of 2.6: 1 BAU/mL = AU/mL × 2.6) [4,5]. Serum samples with IgG titers ≥2080 BAU/mL were diluted using the LIAISON^®^ TrimericS IgG Diluent Accessory by the manufacturer’s recommended dilution factor of 1:20 and tested again.

### 2.3. Testing for Asymptomatic SARS-CoV-2 Infection

Every 15 days, healthcare workers were analyzed with a rapid antigen test (LIAISON^®^ SARS-CoV-2 Ag, DiaSorin S.p.a., Saluggia VC, Italy), following the manufacturer’s instructions, and, if positive, by an SARS-CoV-2 RNA RT-PCR assay (Simplexa COVID-19 Direct DiaSorin) on the same nasopharyngeal swab. The university staff were assayed for the nucleocapsid-specific IgM and/or IgG antibodies (COVID-19 ELISA IgM and COVID-19 ELISA IgG kits, Diatheva srl, Cartoceto, PU, Italy), following the manufacturer’s instructions, and, if positive, by SARS-CoV-2 RNA RT-PCR (Diatheva COVID-19 PCR kit) on a nasopharyngeal swab.

### 2.4. Statistical Analysis

Categorical data are provided as counts and percentages, and continuous data are provided as a median and interquartile range.

To estimate the rate constant k, the time of negativization (t_neg_) and the half-life of IgG (t_1/2_), a kinetic analysis was conducted [3]. The analysis was based on a first-order kinetics equation to describe the descending phase of the IgG concentration in serum over time. The equation was as follows:*C* = *a* × *exp*(−*kt*) [1](1)
where *C* represents the serum IgG concentration, *t* represents time in days, *k* represents the elimination constant and *a* represents the IgG concentration at the two-month time point. To derive the equation parameters, the following equation was used:*ln*(*C*) = −*kt* + *ln*(*a*) [2](2)

Using linear regression analysis, the equation parameters were determined, which allowed for the calculation of both the half-life of IgG (t_1/2_) and the time of their negativization (t_neg_). The time of negativization refers to the point at which the diagnostic test returns negative values (average regression coefficient = −0.986; range −1, −0.851).

Nonparametric ANOVA (Kruskal Wallis test with Dunn’s multiple comparisons post-test) was used to compare both IgG levels and kinetic parameters among vaccination regimens. Paired Friedman’s test with Dunn’s multiple comparison was used for intra-group comparisons at different time points. The Mann–Whitney U test is used to compare differences between the two groups.

ANCOVA was utilized to examine the relationship between the observed IgG titers or the predicted kinetic variables, as dependent variable, with the different vaccine groups, as an explanatory variable, adjusting for other demographic and clinical variables. Tukey’s Least Significant Difference (LSD) test was used as a post hoc test to compare the estimated marginal means between the groups.

Statistical significance was assumed if *p* values were below 0.05. All the analyses were performed using SPSS 23.0 software (SPSS Inc., Chicago, IL, USA). GraphPad Prism (version 8.4.2, GraphPad Software, San Diego, CA, USA) was used to draw the box and whisker plots.

## 3. Results

### 3.1. Longitudinal Analysis of SARS-CoV-2 Anti-Trimeric Spike IgG Levels at Two, Four and Six Months Post-Vaccination

All subjects developed a positive SARS-CoV-2 anti-trimeric Spike IgG antibody response during the follow-up period, with a few exceptions, especially in the ChAd/ChAd group at the four- (8%) and six-month (20%) time points (Table 2). A significant proportion of participants who received ChAd/BNT and BNT/BNT vaccines (52% and 30%, respectively) had a serological test result above the upper limit of quantification (>2080 BAU/mL) at the two-month time point. However, these levels decreased over time throughout the study period (Table 2).

Longitudinal analysis of antibody levels for each group after the second immunization showed that, at each time point, the heterologous vaccination (ChAd/BNT) induced a significantly higher anti-trimeric SARS-CoV-2 Spike protein IgG response than the homologous adenovirus-based vaccine (ChAd/ChAd) (*p* ≤ 0.0001) (Figure 1A, Appendix A, Table 3A). The humoral response induced by ChAd/BNT vaccination was significantly higher than that induced by the homologous mRNA vaccination (BNT/BNT) at the 2-month time point (*p* = 0.0013). However, this difference was less pronounced at the 4-month time point (*p* = 0.0446) and not statistically significant at the 6-month time point. The humoral response induced by the BNT/BNT vaccination, at all times, was significantly higher than the response induced by the ChAd/ChAd vaccination (*p* ≤ 0.0001) (Figure 1A, Table 3A). Intra-group comparisons showed that, for all vaccination groups, the antibody concentration declined over time. For both ChAd/ChAd and ChAd/BNT vaccination, the two-month time point showed significantly higher IgG titers than the four- and six-month time points (both *p* ≤ 0.0001), and the four-month time point showed significantly higher IgG titers than the six-month time point (*p* ≤ 0.01). For BNT/BNT vaccination, the two-month time point showed significantly higher IgG titers than the four- and six-month time points (both *p* ≤ 0.0001), and the four-month time point showed significantly higher IgG titers than the six-month time point (*p* ≤ 0.0001) (Appendix A, Table 3A).

Furthermore, we performed a second analysis of IgG responses, focusing on patients who completed the post-vaccination follow-up and whose serum samples showed a descending phase of the anti-S IgG concentration. This subgroup (ChAd/ChAd (*n* = 124), BNT/BNT (*n* = 420) and ChAd/BNT (*n* = 41)) will be later used for kinetic analysis. The longitudinal analysis of antibody levels for each group, at different time points after the second immunization, showed that, at each time point, the ChAd/BNT vaccination induced a significantly higher SARS-CoV-2 anti-trimeric Spike protein IgG response compared to both BNT/BNT and ChAd/ChAd immunization (*p* ≤ 0.0001). The BNT/BNT vaccination induced higher IgG levels than the ChAd/ChAd vaccination at all time points (*p* ≤ 0.0001) (Figure 1B, Table 3B). Intra-group comparisons showed that, for all vaccination groups, the antibody concentration declined significantly over time (*p* ≤ 0.0001) (Appendix A, Table 3B).

### 3.2. Kinetic Analysis of IgG Concentration Decay following Immunization: Determination of Rate Constant, Half-Life and Time to Negativization

For the subset of 585 patient who completed the post-vaccination follow-up and showed a descending phase of the anti-S IgG concentration, we then analyzed the antibody decay associated with each vaccination strategy (BNT/BNT, ChAd/BNT, ChAd/ChAd) by applying a first-order kinetics equation [3]. Three parameters were evaluated: (1) the rate constant (k), which describes the decay of the IgG concentration over time; (2) the half-life (t_1/2_), which indicates the time required for the IgG concentration to decrease to half of its initial value and (3) the time to negativization (t_neg_), which denotes the time required for the IgG concentration to decrease to a threshold delimited by the limit of detection of the assay (Figure 2).

The heterologous immunization with ChAd/BNT, which produced the highest IgG titer in the interval of 2–6 months after vaccination (Figure 1B, Table 3B), was predicted to have the highest t_neg_ (median [IQR] 372 (304–511) days), significantly higher than that of both the immunization with the homologous BNT/BNT (median [IQR] 300 (242–383) days) (*p* ≤ 0.01) and the immunization with the homologous ChAd/ChAd (median [IQR] 197 (137–292) days) (*p* ≤ 0.0001) (Figure 2A). The BNT/BNT immunization was predicted to induce significantly higher t_neg_ than the ChAd/ChAd immunization (*p* ≤ 0.0001).

When comparing the decay of the IgG concentration between groups, BNT/BNT immunization (median [IQR] 0.0118 (0.0093–0.0141) days^−1^) was predicted to induce a k significantly higher than that induced by the ChAd/ChAd (median [IQR] 0.0099 (0.0077–0.0122) days^−1^) (*p* ≤ 0.0001) and comparable to that induced by the heterologous ChAd/BNT immunization (median [IQR] 0.0105 (0.0088–0.0133) days^−1^) (Figure 2B).

Accordingly, the IgG half-life estimation, which is inversely related to k, showed that ChAd/ChAd immunization (median [IQR] 70 (57–90) days) was predicted to induce a significantly higher t_1/2_ than BNT/BNT (median [IQR] 59 (49–75) days) (*p* ≤ 0.0001) and one that is comparable to that induced by the heterologous ChAd/BNT immunization (median [IQR] 66 (52–79) days) (Figure 2C).

### 3.3. SARS-CoV-2 Antigen and Anti-N Antibodies Testing Results

To monitor for the possible contact of vaccinated subjects with the virus over the course of the study, the healthcare workers and the university staff were periodically tested with a rapid antigen test and anti-nucleocapsid (N) antibodies test, respectively. None of the healthcare workers tested positive for the antigen test during the follow-up. Anti-N IgM and IgG antibodies were identified in university personnel during the analysis (9% ChAd/ChAd, 20% BNT/BNT and 10% ChAd/BNT) (Appendix A). All subjects who tested positive for IgM or IgG anti-N had a negative result for the SARS-CoV-2 RNA RT-PCR test.

### 3.4. Factors Affecting IgG Response

For the subgroup of patients whose IgG decay kinetics were analyzed (*n* = 585), we also used ANCOVA [6] to examine the relationship between the observed IgG titers and the predicted kinetic variables, respectively, as dependent variables, with the vaccine group as an explanatory variable, adjusting for age, sex and vaccine schedule (Table 4 and Appendix A).

The vaccine schedule was predicted by ANCOVA analysis to have a significant impact on all the dependent variables tested (anti-S IgG concentration and kinetic parameters) (*p* = 0.002 for t_neg_; *p* ≤ 0.0001 for all the other variables). Sex and age were not significant factors for any of the variables tested.

The results of the pairwise comparisons predicted that the ChAd/BNT vaccination had a significantly higher effect on the anti-S IgG antibody level at all time points compared to both the BNT/BNT and ChAd/ChAd schedules (*p* ≤ 0.0001). In addition, the BNT/BNT schedule was predicted to have a higher effect on the anti-S IgG antibody level than the ChAd/ChAd schedule at all time points (*p* ≤ 0.0001).

For the kinetic parameters, the IgG induced by the ChAd/BNT vaccination were predicted to have a significantly higher t_neg_ compared to both the BNT/BNT (*p* = 0.003) and ChAd/ChAd (*p* ≤ 0.0001) vaccinations. As for the rate constant, the BNT/BNT vaccination was associated with higher k values than the ChAd/ChAd vaccination (*p* ≤ 0.0001). Conversely, for the parameter t_1/2_, the IgG induced by the ChAd/ChAd vaccination were found to have a significantly higher half-life compared to those induced by the BNT/BNT vaccination (*p* ≤ 0.0001).

Furthermore, we were able to collect more information, in relation to BMI and smoking status, from a subgroup of patients who had filled out a questionnaire within the informed consent form (*n* = 171: 70% ChAd/ChAd (*n* = 120); 6% BNT/BNT (*n* = 10); 24% ChAd/BNT (*n* = 41)). This allowed us to perform a second ANCOVA analysis (Table 5, Appendix A).

This second ANCOVA analysis confirmed that the vaccine schedule has a statistically significant effect on the anti-S IgG antibody level induced at all time points (*p* ≤ 0.0001). However, this time, a not statistically significant effect on the kinetic parameters was associated with vaccination (Table 5). Age and smoking were not predicted to have a statistically significant effect on any of the parameters. Regarding demographic factors, sex was predicted to have a statistically significant effect on the rate constant of antibody decay (*p* = 0.018), with females showing higher k values than males (*p* = 0.0316 by the Mann–Whitney test). BMI was predicted to have a statistically significant effect on the kinetic parameters t_neg_ (*p* = 0.008), k (*p* = 0.009) and t_1/2_ (*p* = 0.008). Therefore, we performed a comparison between patients with a BMI below and above 25, which is considered the threshold between normal weight and overweight. The results showed that patients with a BMI below 25 had higher t_1/2_ and t_neg_ (*p* = 0.0102 by the Mann–Whitney test) and lower k values, indicating a slower rate of antibody decay compared to patients with a BMI above 25.

In line with the previous ANCOVA analysis (Table 4), both ChAd/BNT and BNT/BNT schedules were predicted to have a significant positive effect on the anti-S IgG antibody level at all time points (two, four and six months) compared to the ChAd/ChAd schedule (*p* ≤ 0.0001). ChAd/BNT vaccination was also associated with higher t_neg_ when compared to the ChAd/ChAd schedule (*p* = 0.042).

## 4. Discussion

As of April 2023, eleven vaccines have received a WHO Emergency Use Listing (EUL) [7], in addition to many others authorized by one or more regulatory authorities for specific or widespread use [8].

Heterologous vaccination is expected to play an increasingly important role in the global COVID-19 vaccine strategy due to several factors, including the needs to optimize the effectiveness of available vaccines, respond to emerging variants of the virus and address variable vaccine supply. While more research is needed to fully understand the potential benefits and risks of heterologous schedules, initial studies have been promising, and this approach is likely to become an important tool in the ongoing fight against COVID-19 and other infectious diseases [2,9,10,11].

In this study, we analyzed the humoral response of individuals vaccinated with either homologous mRNA vaccine (BNT/BNT), homologous adenovirus-based vaccine (ChAd/ChAd) or heterologous (ChAd/BNT) vaccine over a six-month period following vaccination. As previously reported [2], safety considerations associated with the ChAdOx1-S vaccine have prompted some European countries, including Italy, to recommend a switch from a homologous booster to a heterologous booster, such as BNT162b2. In our previous work, we reported on the analysis of anti-S response in the serum of vaccinated subjects two months after vaccination [2]. In this work, we expanded our analysis, including the four- and six-month time points, to enable a longitudinal evaluation of the humoral response associated with the different vaccine regimens.

The main limitations of the study are the relatively small sample size for the ChAd/ChAd and ChAd/BNT groups compared to the BNT/BNT group, which may limit the statistical power to detect significant differences between the groups. Additionally, the study only evaluated the humoral response to the vaccines and did not investigate the cellular immune response or the effectiveness of the vaccines in preventing infection or disease. While our results were generated through a single SARS-CoV-2 TrimericS IgG antibody assay, this method has shown a high concordance with neutralizing IgG antibodies [4,5].

Like prior investigations conducted on Italian cohorts in real-world settings [12,13,14,15], our study, focused on a cohort of healthy workers, predominantly in young to middle adulthood and belonging to the medical and academic settings, reveals that the vaccination is remarkably effective. All subjects developed a positive SARS-CoV-2 TrimericS IgG antibody response during the follow-up period, with a few exceptions, especially in the ChAd/ChAd group at 4 and 6 months. A significant proportion of participants who received ChAd/BNT and BNT/BNT vaccines (52% and 30%, respectively) had antibody levels above the upper limit of quantification at the two-month time point.

The anti-trimeric SARS-CoV-2 Spike protein IgG significantly declined for all three vaccination groups over time (2–6 months period), but the decrease in IgG titers was more pronounced in the BNT/BNT vaccination group compared to that in the ChAd/ChAd and ChAd/BNT vaccination groups.

When comparing the schedules, the heterologous ChAd/BNT vaccination induced a significantly higher anti-trimeric SARS-CoV-2 Spike protein IgG response than both the ChAd/ChAd and BNT/BNT vaccinations at each time point. However, differences between ChAd/BNT and BNT/BNT vaccination decreased from the two-month to the four-month time point and became statistically non-significant at the six-month time point. This indicates that the combination of the two different vaccine technologies resulted in a more robust or at least equal immune response than homologous vaccination, in line with several previous findings [2,9,10,11]. In particular, a recent comprehensive review on the safety, immunogenicity, and effectiveness of heterologous vaccine schedules indicated that, overall, vectored vaccines have shown enhanced immunogenicity when administered before or after mRNA compared to homologous vectored vaccine schedules [16].

Moreover, a subgroup of participants whose serum samples showed a descending phase of the anti-S IgG concentration throughout the investigation was further analyzed for kinetic studies. The heterologous ChAd/BNT vaccination is associated with the highest time to negativization of anti-S IgG (372 days), followed by BNT/BNT (300 days) and ChAd/ChAd (197 days) vaccination, respectively. However, the IgG half-life estimation, which is inversely related to the rate constant, showed that ChAd/ChAd immunization resulted in a significantly higher half-life of IgG compared to BNT/BNT immunization. The half-life of IgG was comparable between ChAd/BNT and ChAd/ChAd immunization. Overall, these results suggest that heterologous ChAd/BNT immunization could offer longer-lasting protection against SARS-CoV-2 compared to homologous vaccination strategies, possibly due to a combination of the higher IgG induced by vaccination and a slower decay of the IgG concentration over time. When comparing the two homologous vaccination strategies, BNT/BNT immunization, which induced higher IgG responses than ChAd/ChAd, was associated with a higher decay of the IgG titer over time, for reasons currently unknown. It is important to note that these results are based only on the subgroup of patients who demonstrated a declining phase of the anti-S IgG concentration. Further studies are required to confirm these findings at later time points and in larger populations.

When analyzing for clinical variables influencing the immune response, we found that only the vaccine schedule had a significant impact on both IgG titers and kinetic parameters, while sex and age were not significant factors. ChAd/BNT vaccination was predicted to elicit the highest anti-S IgG antibody response and was associated with the lowest time to negativization. A more defined subset was also analyzed for BMI and smoking status, showing that IgG from patients with a BMI above 25 (overweight) were associated with a lower half-life and time to negativization. Obesity is known to negatively affect vaccination responses, and the potential association between obesity and a reduced effectiveness of COVID-19 vaccine-induced neutralizing humoral immunity has been previously observed [17]. These findings highlight the importance of considering the vaccine schedule and individual factors, such as BMI, in optimizing vaccination strategies for COVID-19.

In summary, our findings suggest that the ChAd/BNT heterologous vaccination elicits a stronger immune response than the BNT/BNT homologous vaccination, while the BNT/BNT homologous vaccination induces a stronger immune response than the ChAd/ChAd homologous vaccination. Understanding the long-term kinetics of antibodies to SARS-CoV-2 vaccination or exposure, as well as individual characteristics influencing it, is crucial to comprehending protective immunity against COVID-19 and developing effective surveillance strategies. These results underscore the importance of the ongoing monitoring of vaccine effectiveness and the necessity of booster shots in order to sustain immunity against SARS-CoV-2. In conclusion, this study provides important information on the humoral response to SARS-CoV-2 vaccines that has important implications for public health policy and vaccine development.

## Figures and Tables

**Figure 1 viruses-15-01162-f001:**
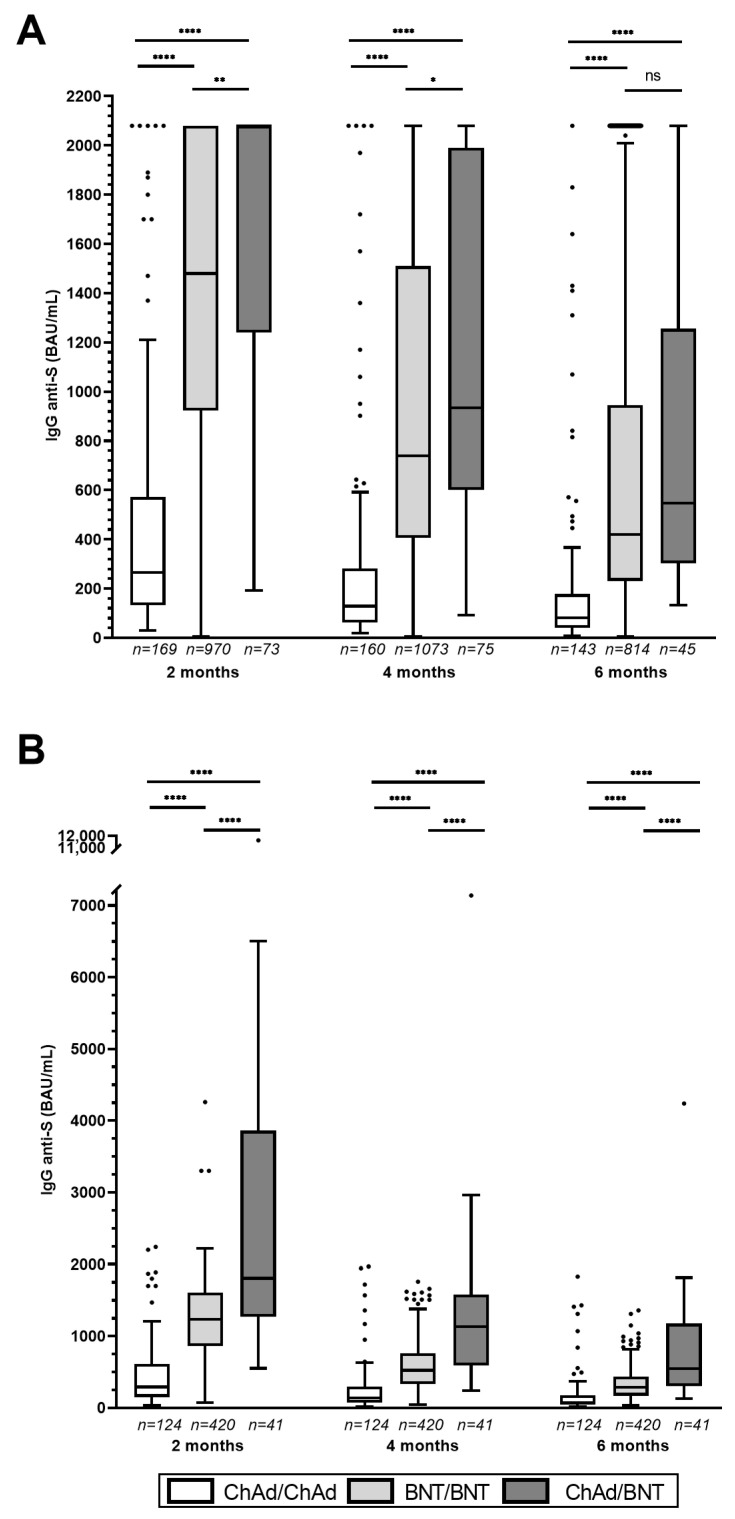
Inter-group comparison of SARS-CoV-2 anti-trimeric Spike protein IgG among three different groups of immunized subjects at two, four and six months after vaccination. (**A**) All subjects (different number of patients reported in the figure). (**B**) Subgroup of subjects who completed the post-vaccination follow-up and whose blood samples showed a descending phase of the anti-S IgG concentration (*n* = 585). The box represents the interquartile range, and the whiskers extend to the lowest and highest value within 1.5 times the interquartile range from the hinges, respectively (Tukey-style). Kruskal Wallis test with Dunn’s post hoc multiple comparisons. * *p* ≤ 0.05; ** *p* ≤ 0.01; **** *p* ≤ 0.0001; ns = not statistically significant.

**Figure 2 viruses-15-01162-f002:**
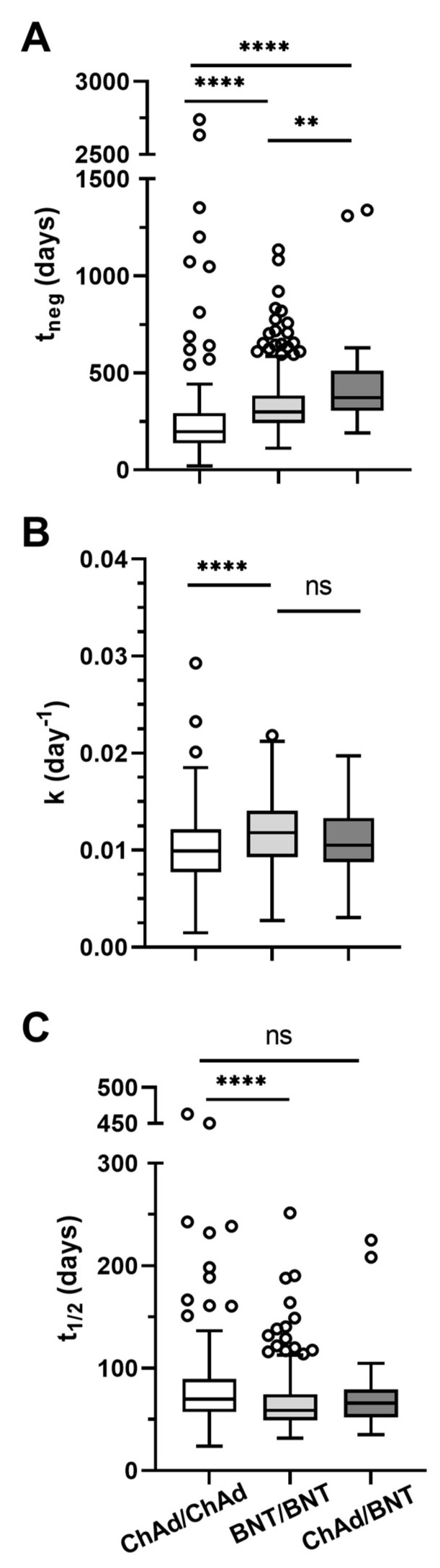
Kinetic analysis of IgG concentration decay following immunization with ChAd/ChAd (*n* = 124), BNT/BNT (*n* = 420) and ChAd/BNT (*n* = 41) among vaccinated subjects using a first-order kinetics equation: (**A**) time of negativization (t_neg_), (**B**) rate constant k, (**C**) half-life (t_1/2_). The box-whiskers are presented in Tukey style. Kruskal Wallis test with Dunn’s post hoc multiple comparisons. **** *p* ≤ 0.0001, ** *p* ≤ 0.01, ns = not statistically significant.

**Table 1 viruses-15-01162-t001:** Characteristics of the three groups of vaccinated subjects.

**A**. All subjects
Vaccine Schedule		**2 Months**	**4 Months**	**6 Months**
All			*n* = 1212	*n* = 1308	*n* = 1002
Gender	Male	399 (33%)	418 (32%)	335 (33%)
Age	Years (median, IQR)	51 (44–58)	51 (44–58)	51 (44–58)
ChAd/ChAd			*n* = 169 (14%)	*n* = 160 (12%)	*n* = 143 (14%)
Gender	Male	75 (44%)	73 (46%)	64 (45%)
Age	Years (median, IQR)	56 (50–62)	57 (49–62)	56 (49–62)
BNT/BNT			*n* = 970 (80%)	*n* = 1073 (82%)	*n* = 814 (81%)
Gender	Male	290 (30%)	309 (29%)	250 (31%)
Age	Years (median, IQR)	50 (42–57)	51 (42–58)	51 (43–57)
ChAd/BNT			*n* = 73 (6%)	*n* = 75 (6%)	*n* = 45 (4%)
Gender	Male	34 (47%)	36 (48%)	21 (47%)
Age	Years (median, IQR)	53 (47–58)	53 (47–58)	55 (45–59)

**B.** Subjects who completed the post-vaccination follow-up and had a descending phase of the anti-S IgG concentration
	**All**	**ChAd/ChAd**	**BNT/BNT**	**ChAd/BNT**
*n* = 585	*n* = 124 (21%)	*n* = 420 (72%)	*n* = 41 (7%)
Gender	Male	204 (35%)	55 (44%)	130 (31%)	19 (46%)
Age	Years (median, IQR)	52 (45–59)	56 (49–62)	51 (43–58)	53 (42–58)

ChAd/ChAd denotes a ChAdOx1 nCoV-19 (ChAd) COVID-19 vaccine (Vaxzevria, AstraZeneca) for prime and boost doses. BNT/BNT denotes BNT162b2 (BNT) COVID-19 vaccine (Comirnaty, Pfizer–BioNTech) for prime and boost doses. ChAd/BNT denotes a ChAd vaccine for a prime dose and a BNT vaccine for a boost dose. IQR: interquartile range.

**Table 2 viruses-15-01162-t002:** Number of subjects below the cut-off for positivity (33.8 BAU/mL) or above the upper limit of quantification (2080 BAU/mL) in the different groups of those vaccinated during the study.

Vaccine Schedule	<33.8 BAU/mL			>2080 BAU/mL		
	2 Months	4 Months	6 Months	2 Months	4 Months	6 Months
ChAd/ChAd	1/169 (0.6%)	12/160 (8%)	28/143 (20%)	5/169 (3%)	4/160 (3%)	3/143 (2%)
BNT/BNT	4/970 (0.4%)	3/1073 (0.3%)	7/814 (0.9%)	288/970 (30%)	184/1073 (17%)	112/814 (14%)
ChAd/BNT	0/73 (0%)	0/75 (0%)	0/45 (0%)	38/73 (52%)	18/75 (24%)	3/45 (7%)
Total	5/1212 (0.4%)	15/1308 (1%)	35/1002 (3%)	331/1212 (27%)	206/1308 (16%)	118/1002 (12%)

**Table 3 viruses-15-01162-t003:** SARS-CoV-2 anti-trimeric Spike protein IgG levels and kinetic parameters among three different groups of immunized subjects.

**A**. All Subjects
Experimental parameter	IgG titer (BAU/mL)		**ChAd/ChAd**	**BNT/BNT**	**ChAd/BNT**
2 monthsMedian (IQR)	*n* = 169 (14%)265(133–573)	*n* = 970 (80%)1480(923–>2080)	*n* = 73 (6%)>2080(1240–>2080)
4 monthsMedian (IQR)	*n* = 160 (12%)129(62–283)	*n* = 1073 (82%)739(407–1510)	*n* = 75 (6%)934(601–1990)
6 monthsMedian (IQR)	*n* = 143 (14%)80(40–178)	*n* = 814 (81%)419(231–946)	*n* = 45 (5%)548(302–1255)

**B**. Subjects who completed the post-vaccination follow-up and with descending phase of the anti-S IgG concentration.
			**ChAd/ChAd***n* = 124 (21%)	**BNT/BNT***n* = 420 (72%)	**ChAd/BNT***n* = 41 (7%)
Experimental parameter	IgG titer (BAU/mL)	2 monthsMedian (IQR)	291(149–614)	1235(858–1608)	1806(1270–3860)
4 monthsMedian (IQR)	139(69–297)	525(331–763)	1130(590–1582)
6 monthsMedian (IQR)	74(40–174)	288(169–433)	548(302–1176)
Kinetic parameter *	t_neg_(days)	Median (IQR)	197(137–292)	300(242–383)	372(304–511)
k(days^−1^)	Median (IQR)	0.0099(0.0077–0.0122)	0.0118(0.0093–0.0141)	0.0105(0.0088–0.0133)
t_1/2_ (days)	Median (IQR)	70(57–90)	59(49–75)	66(52–79)

ChAd/ChAd denotes a ChAdOx1 nCoV-19 (ChAd) COVID-19 vaccine (Vaxzevria, AstraZeneca) for prime and boost doses. BNT/BNT denotes BNT162b2 (BNT) COVID-19 vaccine (Comirnaty, Pfizer–BioNTech) for prime and boost doses. ChAd/BNT denotes a ChAd vaccine for the prime dose and a BNT vaccine for the boost dose. k = rate constants, t_1/2_ = half-life, t_neg_ = time to negativization. * Kinetic parameters were derived by applying a first-order kinetic equation.

**Table 4 viruses-15-01162-t004:** *p* values resulting from the analysis of covariance (ANCOVA) in a group of 585 vaccinated subjects: anti-S IgG antibody level at 2, 4 and 6 months after vaccination and kinetic parameters, corrected for vaccine schedule, sex and age.

Factor	Anti-S IgG Antibody Level at			Kinetic Parameters		
	2 Months	4 Months	6 Months	t_neg_	K	t_1/2_
Vaccine schedule	≤0.0001	≤0.0001	≤0.0001	0.002	≤0.0001	≤0.0001
Sex	0.160	0.943	0.577	0.927	0.366	0.540
Age	0.794	0.469	0.631	0.322	0.415	0.450
Pairwise comparisons *	ChAd/BNT > BNT/BNT (≤0.0001)	ChAd/BNT > BNT/BNT (≤0.0001)	ChAd/BNT > BNT/BNT (≤0.0001)	ChAd/BNT > BNT/BNT (0.003)	BNT/BNT > ChAd/ChAd (≤0.0001)	ChAd/ChAd > BNT/BNT (≤0.0001)
	ChAd/BNT > ChAd/ChAd (≤0.0001)	ChAd/BNT > ChAd/ChAd (≤0.0001)	ChAd/BNT > ChAd/ChAd (≤0.0001)	ChAd/BNT > ChAd/ChAd (≤0.0001)		
	BNT/BNT > ChAd/ChAd (≤0.0001)	BNT/BNT > ChAd/ChAd (≤0.0001)	BNT/BNT > ChAd/ChAd (≤0.0001)			

* Pairwise comparisons are calculated by Tukey’s LSD post hoc analysis. Only statistically significant differences are reported.

**Table 5 viruses-15-01162-t005:** Results of analysis of covariance (ANCOVA) in a subgroup of 171 subjects: anti-S IgG antibody level at 2, 4 and 6 months after vaccination and kinetic parameters, corrected for several factors.

Factor	Anti-S IgG Antibody Level at			Kinetic Parameters		
	2 Months	4 Months	6 Months	t_neg_	K	t_1/2_
Vaccine schedule	≤0.0001	≤0.0001	≤0.0001	0.068	0.366	0.293
Sex	0.288	0.536	0.565	0.197	0.018	0.082
Age	0.101	0.125	0.084	0.990	0.763	0.686
Smoking	0.811	0.785	0.875	0.976	0.958	0.750
BMI	0.843	0.091	0.044	0.008	0.009	0.008
Pairwise comparisons *	ChAd/BNT > ChAd/ChAd (≤0.0001)	ChAd/BNT > ChAd/ChAd (≤0.0001)	ChAd/BNT > ChAd/ChAd (≤0.0001)	ChAd/BNT > ChAd/ChAd (0.042)		
	BNT/BNT > ChAd/ChAd (≤0.0001)	BNT/BNT > ChAd/ChAd (≤0.0001)	BNT/BNT > ChAd/ChAd (≤0.0001)			

* Pairwise comparisons are calculated by Tukey’s LSD post hoc analysis. Only statistically significant differences are reported.

## Data Availability

Not applicable.

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
