# Peer review of "Comparing Heterologous and Homologous COVID-19 Vaccination: A Longitudinal Study of Antibody Decay"

_viruses, 2023, doi:10.3390/v15051162_

Round 1

Reviewer 1 Report

Major concerns.

1. The introduction in the first paragraph must be the background of the SARS-CoV-2 and COVID-19 vaccine, in general, to emphasise how important disease and vaccine prevention are. Moreover, the introduction should focus on the homologous and heterologous regimens.
But the first paragraph of this manuscript seems to be a materials and methods section.

Recommend re-write the introduction following the IMRaD structure.

2. It would be more informative If the graph was presented as scatter or dot plots. The individual dot could be more interesting than the bar and lines with extreme values.

Minor concerns.

1. Suggest adding the conversion factor of the TrimericS because of the result from this instrument report as AU/mL to make it clearer. Not all readers know the unit of each instrument.

Comments.

1. Suggest using "ChAd" instead of "Chad" to make it consistent throughout the manuscript.

2. Have you tried to create the XY plot between time (x-axis) and antibody levels (y-axis)?

Using these figures with the result would be great, especially the long-term waning trend, half-life and time to negativisation.

Reviewer 2 Report

The study is very well designed. A population of healthcare workers was used and the results obtained suggest that heterologous vaccination provokes a stronger immune response than homologous vaccination with a greater persistence of antibodies. The study was done  evaluating both IgG and IgM response. The study was done by chemiluminescence. Interestingly, a follow-up of the immune response after different types of vaccines was done in a population of different ages and also an evaluation of the immune response . The understanding of the kinetics of antibody production after vaccination or exposure to SARS-COV-19 as well as individual characteristics of the individual and factors influencing either the disease or vaccination is very important for understanding protective immunity. The study is one of the first to address the aspect in search of important information on the humoral response to SARS-CoV-2 vaccines, with implications on different aspects

Author Response

Thank you the reviewer for careful and thorough reading of this manuscript and for the thoughtful comments.

Round 2

Reviewer 1 Report

After revision, the manuscript is acceptable for publication.

For the visualisation. It depended on the author's decision. I suggest a better appearance.

Your data seems to have a log-normal distribution.

Have you tried to plot with the log y-axis, or log-transformed?